# *Salmonella*-Based Therapy Targeting Indoleamine 2,3-Dioxygenase Restructures the Immune Contexture to Improve Checkpoint Blockade Efficacy

**DOI:** 10.3390/biomedicines8120617

**Published:** 2020-12-16

**Authors:** Nancy D. Ebelt, Edith Zuniga, Monica Marzagalli, Vic Zamloot, Bruce R. Blazar, Ravi Salgia, Edwin R. Manuel

**Affiliations:** 1Department of Immuno-Oncology, Beckman Research Institute of the City of Hope, Duarte, CA 91010, USA; nebelt@coh.org (N.D.E.); edzuniga@coh.org (E.Z.); monica.marzagalli@unimi.it (M.M.); vzamloot@coh.org (V.Z.); 2Department of Pediatrics, Division of Blood and Bone Marrow Transplantation, University of Minnesota Medical School, Minneapolis, MN 55455, USA; blaza001@umn.edu; 3Department of Medical Oncology and Therapeutics Research, City of Hope National Medical Center, Duarte, CA 91010, USA; rsalgia@coh.org

**Keywords:** non-small cell lung cancer, immune checkpoint blockade, anti-PD-1, anti-CTLA4, *Salmonella typhimurium*, small-hairpin RNA, indoleamine 2,3-dioxygenase

## Abstract

Therapeutic options for non-small cell lung cancer (NSCLC) treatment have changed dramatically in recent years with the advent of novel immunotherapeutic approaches. Among these, immune checkpoint blockade (ICB) using monoclonal antibodies has shown tremendous promise in approximately 20% of patients. In order to better predict patients that will respond to ICB treatment, biomarkers such as tumor-associated CD8^+^ T cell frequency, tumor checkpoint protein status and mutational burden have been utilized, however, with mixed success. In this study, we hypothesized that significantly altering the suppressive tumor immune landscape in NSCLC could potentially improve ICB efficacy. Using sub-therapeutic doses of our *Salmonella typhimurium*-based therapy targeting the suppressive molecule indoleamine 2,3-dioxygenase (shIDO-ST) in tumor-bearing mice, we observed dramatic changes in immune subset phenotypes that included increases in antigen presentation markers, decreased regulatory T cell frequency and overall reduced checkpoint protein expression. Combination shIDO-ST treatment with anti-PD-1/CTLA-4 antibodies enhanced tumor growth control, compared to either treatment alone, which was associated with significant intratumoral infiltration by CD8^+^ and CD4^+^ T cells. Ultimately, we show that increases in antigen presentation markers and infiltration by T cells is correlated with significantly increased survival in NSCLC patients. These results suggest that the success of ICB therapy may be more accurately predicted by taking into account multiple factors such as potential for antigen presentation and immune subset repertoire in addition to markers already being considered. Alternatively, combination treatment with agents such as shIDO-ST could be used to create a more conducive tumor microenvironment for improving responses to ICB.

## 1. Introduction

Non-small cell lung cancer (NSCLC) tumors contain large percentages of T cells (25–46% of the CD45^+^ fraction), many expressing tumor antigen-specific T cell receptors (TCRs) [1,2]. However, most of these T cells lack effector function and are prone to activation-induced apoptosis due to chronic stimulation by tumor-derived antigens [3,4]. An important marker of these dysfunctional T cells is programed death-1 (PD-1), a cell surface receptor that can be activated to cause T cell apoptosis by tumor cells or other immune cells expressing programed death-ligand 1 (PD-L1) [5,6,7]. Antibody-mediated inhibition of the PD-1/PD-L1 axis is a promising new treatment option for non-small cell lung cancer (NSCLC), which continues to be the leading cause of cancer-related death in the United States and worldwide [8,9,10]. Several anti-PD-1/PD-L1 antibodies, including pembrolizumab, nivolumab, and atezolizumab, have been approved for the treatment of advanced NSCLC after remarkable improvement in median overall NSCLC patient survival was demonstrated in clinical trials [11,12].

The presence of another cell surface molecule, cytotoxic T lymphocyte-associated protein 4 (CTLA-4), suppresses T cell responses by acting as a homolog for CD28 that competes for binding to activating MHC co-receptors present on antigen presenting cells (APCs) [13]. Treatment of NSCLC patients with antibodies targeting CTLA-4 (e.g., ipilumumab) in conjunction with chemotherapy results in modest increases in immune-related, progression-free survival, but overall survival is unchanged [14]. Greater benefit was seen with the combination of ipilimumab and nivolumab, which showed significantly improved overall survival compared to chemotherapy and nivolumab alone [15], resulting in recent FDA approval for frontline treatment of metastatic NSCLC with the ipilimumab and nivolumab combination. These cell surface molecules, and a few others, constitute the immune “checkpoint” proteins. Although the magnitude and durability of responses induced by immune checkpoint blockade (ICB) are unrivaled by current treatment options, only a small percentage of cancer patients will experience clinical benefit [16,17], indicating mechanisms of resistance to checkpoint blockade that must be alleviated before responses can be improved.

Two major cell types have been connected to checkpoint blockade failure: regulatory CD4^+^ T cells (Tregs) and myeloid derived suppressor cells (MDSCs) [18,19,20]. Tregs and MDSCs contribute to a suppressive microenvironment through distinct and overlapping mechanisms. Tregs, found in tumor tissue and draining lymph nodes, have high expression of CTLA-4 and Lag-3 that facilitate suppression of dendritic cells (DCs) and T cells through direct cell-to-cell contact. They also secrete immunosuppressive cytokines such as IL-10, transforming growth factor β (TGF-β), and IL-35, which downregulate IFNγ responses, inhibit antigen presentation by APCs, and impair T cell proliferation [21]. MDSCs suppress T cell activation and proliferation through upregulation of reactive oxygen species (ROS) and arginase, as well as indoleamine 2,3-dioxygenase (IDO) [22,23,24] which suppresses T-cell function by converting essential tryptophan into suppressive kynurenines [25]. MDSCs can also suppress NK cytotoxic function [26], inhibit antigen presentation by APCs, and induce formation of Tregs through secretion of IL-10 [27,28]. Finally, MDSCs express significant levels of PD-L1, PD-1, and CTLA-4 that can also interfere with T cell activation [29].

Recently, a novel subset of anti-tumor neutrophils have been identified in the early stages of NSCLC that are capable of inducing anti-tumor immunity through attainment of antigen presentation capabilities [30,31,32,33]. Our recent studies in pancreatic ductal adenocarcinoma (PDAC) have also shown that depletion of neutrophils and monocytes abrogate the anti-tumor effects of anti-PD-1 treatment, demonstrating the importance of tumor-associated myeloid cells in checkpoint blockade efficacy [34]. Increasing antigen stimulation of T cells through increased presentation by APCs and/or converted MDSC may sensitize permanently dysfunctional T cells to ICB therapy. Significantly decreasing the frequency of various immunosuppressive subsets may also be necessary to create a tumor microenvironment (TME) more conducive for ICB.

Utilizing various cancer models, we have confirmed that our *Salmonella typhimurium*-based platform targeting IDO (shIDO-ST) is engulfed by tumor-infiltrating neutrophils, leading to hyperactivation of their cytotoxic activity and subsequent tumor regression [22,35]. In this study, we hypothesized that shIDO-ST treatment might improve the efficacy of ICB treatment in NSCLC if a greater breadth of anti-tumor immune responses could be generated including increasing antigen presentation by various immune subsets. While we confirmed the ability of high-dose shIDO-ST treatment to produce neutrophils with APC function, the overall immune response was highly skewed to this cytotoxic subset and not conducive for improving ICB efficacy [36,37]. Alternatively, we discovered that sub-therapeutic doses of shIDO-ST could prevent the dominant neutrophil response and augment antigen presentation in professional and non-conventional APCs including dendritic cells, macrophages, and monocytes. Additionally, low-dose shIDO-ST treatment reduced the frequency of suppressive Tregs and expression of immune checkpoint molecules, ultimately improving efficacy of ICB therapy in an aggressive model of NSCLC. Our ability to demonstrate synergy between shIDO-ST and ICB treatment may support their combination as a strategy to enhance response rates in NSCLC patients [38,39].

## 2. Experimental Section

### 2.1. Animals and Cell Lines

Experiments involving animals were performed according to the standards of the Institution Animal Care and Use Committee (IACUC) at the City of Hope (IACUC protocol 17128; approved 6 February 2018)). Male C57BL/6 mice, 8–10 weeks old, were obtained from breeding colonies housed at the City of Hope Animal Research Center. *Kras^LSL-G12D/+^; Trp53^R270H/+^* (KP) mice were provided by Dr. Thomas Ludwig (Ohio State University, Columbus, OH, USA) [40]. The Lewis Lung Carcinoma (LLC1) cell line was obtained from ATCC^®^ (CRL-1642). Cells were maintained in RPMI media containing 10% FBS, 2mM l-glutamine and pen/strep. Prior to subcutaneous implantation into C57BL/6 mice, cells were passaged ≤5 times and maintained at ≤80% confluency.

### 2.2. Salmonella Typhimurium (ST)

YS1646 was obtained from ATCC^®^ (202165™). YS1646 was cultured in modified LB media containing MgSO_4_ and CaCl_2_ in place of NaCl. The pAKlux2 plasmid was a kind gift from Attila Karsi (Department of Basic Sciences, College of Veterinary Medicine, Mississippi State University, Mississippi State, MS, USA; Addgene #14080). ShScr and shIDO plasmids (Sigma-Aldrich, Carlsbad, CA, USA) [35] were electroporated into ST strains using a BTX electroporator (1 mm gap cuvettes, settings: 1.8 kV, 186 ohms), spread onto LB plates containing 100 µg/mL ampicillin and incubated overnight at 37 °C.

### 2.3. ST Administration and Neutrophil Isolation in Tumor-Free Mice

For blood and spleen neutrophil isolations, on the first day mice were administered 5 × 10^6^ cfu of shScr, shIDO-ST or HBSS volume equivalent followed by 2 × 10^6^ cfu on the second and 1 × 10^6^ cfu on the third consecutive day intravenously via the retro-orbital vein. Blood and spleens were collected after euthanization 48 or 72 h after the third ST administration. For peritoneal neutrophil isolation, 4 × 10^7^ cfu of shScr, shIDO-ST or HBSS volume equivalent was injected into the intraperitoneal cavity of mice followed by 2 × 10^7^ cfu on the second day. Three hours after the second administration, mice were sedated and 5 mL of HBSS was injected into the intraperitoneal cavity; the abdomen was massaged to dislodge cells into the fluid. Then, the HBSS with cells was removed using a large gauge needle. The solution was then overlaid on top of a 0/52/62.5/78 percent Percoll gradient and centrifuged at 700× *g* for 30 min without acceleration or brake. Individual layers were removed by transfer pipette to individual tubes for washing. Wright’s staining (Wright’s Giemsa, WG16-500 mL, Sigma-Aldrich, Carlsbad, CA, USA) was performed on 10–25 µL of washed layers that were smeared and allowed to dry on slides before fixation in 100% methanol.

### 2.4. Establishment of Spontaneous Lung Tumors in KP Mice

Tumor growth in the lungs of KP mice was initiated through inhalation of adenovirus expressing Cre recombinase (Ad5CMVCre, University of Iowa, Iowa City, IA, USA), Lot#Ad4067), according to the intranasal method described in DuPage, et al. 2009 [41]. Intranasal administration of AdCre occurred when the mice were 8 weeks old followed by treatment with ST 65 days later. Lungs were removed for histology 24 h after ST treatment.

### 2.5. Immunohistochemistry (IHC)

IHC was performed on Ventana Discovery Ultra IHC autostainer (Ventana Medical Systems, Roche Diagnostics, Santa Clara, CA USA) according to manufacturer’s protocols. Briefly, tissue samples were sectioned at a thickness of 5 μm and put on positively charged glass slides. Deparaffinization, rehydration, endogenous peroxidase activity inhibition and antigen retrieval were all performed on the automated stainer. Slides were then incubated with primary antibodies, followed by DISCOVERY HQ and DISCOVERY HQ-HRP system, visualized with ChromoMap DAB detection Kit (Ventana Medical Systems, Roche Diagnostics, Santa Clara, CA, USA). The slides were then counterstained with haematoxylin (Ventana Medical Systems, Roche Diagnostics, Santa Clara, CA, USA) and coverslipped. Antibodies used: CD4, CD8a, and CD11c at a dilution of 1:100 (Cell Signaling Technologies). DAB staining out of total nuclei per field was done using ImageJ (Fiji v1.53a, U. S. National Institutes of Health, Bethesda, Maryland, MD, USA). DAB and hematoxylin channels were separated using color deconvolution (H-DAB preset), and thresholds were set to cover positive staining area for DAB and positive staining nuclei for hematoxylin. DAB-positive nuclei were quantified by dividing DAB threshold area by DAB threshold area plus hematoxylin threshold area.

### 2.6. Luminescent Tumor Growth Tracking

The right-side midsections of mice were shaved and LLC1 cells (2 × 10^5^) were implanted subcutaneously in HBSS. Tumors were allowed to grow to an average of ~200 mm^3^ and then injected retro-orbitally with 1 × 10^6^ ST electroporated with the pAKlux2 plasmid. Mice were imaged for luminescence in the Small Animal Imaging Core at City of Hope using the Lago biphotonic imaging system (Spectral Instruments, Tucson, AZ, USA).

### 2.7. Subcutaneous Tumor Growth and Treatment

LLC1 cells were injected subcutaneously (2 × 10^5^ per mouse) in the right-side midsection of the mice. After 6 days of growth, treatment with shScr-ST or shIDO-ST began with three consecutive daily doses of 1 × 10^6^ cfu per mouse. On the first day of ST treatment, mice were concurrently given 200 µg of anti-PD-1 (clone J43) antibody or IgG equivalent (BioXCell, Lebanon, NH, USA) and 75 µg of anti-CTLA-4 (9H10) antibody or IgG equivalent (BioXCell, Lebanon, NH, USA). Treatments continued every three days until the end of the experiment (maximum tumor growth of 15 mm diameter for control groups). Tumor volumes were measured thrice weekly using manual calipers. Lower doses from previously published regimens of anti-PD-1 [34,42] and anti-CTLA-4 [43,44] were chosen to more readily observe potential synergy with shIDO-ST.

### 2.8. Flow Cytometry

One million live cells were counted using trypan blue and first stained with a fixable viability dye (eBiosciences 65-0866-14, Thermo Fisher Scientific, Waltham, MA, USA) for 30 min at 4 °C. Cells were washed in flow wash buffer (PBS with 0.05% sodium azide and 1% FBS) and stained with surface antibodies for 40 min at 4 °C. Cells were washed in flow buffer and fixed in flow buffer plus 1% PFA before filtering through 40 μm mesh strainer/tube (BD Biosciences, San Jose, CA, USA). Flow cytometry was performed on the BD FACSCelesta cytometer and data was analyzed using FlowJo Version 10 (Becton, Dickinson & Co., Franklin Lakes, NJ, USA). For flow cytometry from spleens, spleens were removed and mashed in complete RPMI medium (10% FBS with glutamine and pen/strep) on a 70 µm mesh strainer with a syringe plunger. Cells were washed in PBS and stained using the above protocol. For flow cytometry on tumor cells, tumors were excised and digested mechanically by mincing with a sterile scalpel before digestion in 1 mg/mL collagenase I (Sigma-Aldrich C5138, Carlsbad, CA, USA,) plus 1% FBS for 1.5 h shaking at 200 rpm in a 37 °C incubator. Dissociated tumor cells were spun at 450 g for 10 min and filtered through a 70 μm strainer. After washing, these cells were stained for flow cytometry following the above protocol. Flow cytometry antibodies used from BD biosciences (San Jose, CA, USA): CD45-PerCPCy5.5 (550994), CD8-APC-R700 (564983), CD4-APC-H7 (560246), Ly6G-BV605 (563005), Ly6C-FITC (561085), CD11b-APC (561690), PD-1-BV421 (562584), CTLA-4-PE (553720), CD45-APC-R700 (565478), CD3-PerCPCy5.5 (551071), NK1.1-BV650 (564143), CD11c-BV605 (563057), CD45-BV650 (563410), CD4-BV786 (563331), CD25-PerCPcy5.5 (551071), MHCII-PerCPCy5.5 (562363), CD11c-APC-R700 (565871), IFN-gamma-APC (554413), CD27-BV450 (561245), andCD62L-BV605 (562720). Flow cytometry antibodies used from eBiosciences (Thermo Fisher Scientific; Waltham, MA, USA): F4/80-Super Bright 780 (78-4801-82), PD-L1-Super Bright 645 (64-5982-82), PD-L1-Super Bright 780 (78-5982-80), CD86-Super Bright 645 (64-0862-80), FoxP3-APC (17-5773-82), and Granzyme B-PE (12-8898-80). For intracellular staining against CTLA-4, cells were permeabilized using the Cytofix/Cytoperm Plus kit (555028, BD Biosciences; San Jose, CA, USA). For intracellular staining against FoxP3, cells were permeabilized using the FoxP3/transcription factor staining buffer set from eBiosciences (00-5523-00, Thermo Fisher Scientific; Waltham, MA, USA).

### 2.9. Kaplan-Meier Plots Using Human NSCLC Data

KMplot.com [45] was used to create Kaplan–Meier plots using human data sets. Lung adenocarcinomas (*n* = 513) were selected from the pan-cancer RNA-seq dataset for analysis of overall survival comparing high and low expression of CD11c (ITGAX) and MHCII (averaged expression of HLA-DR, HLA-DRB1, HLA-DRB5, and HLA-DRB6). Three restriction scenarios were used: (1) no restriction of subtypes, no restriction of cellular content, (2) no restriction of subtypes, restriction to samples with enriched CD4 and CD8 content and decreased Treg content, and (3) no restriction of subtypes, restriction to samples with decreased CD4 and CD8 content and enriched Treg content.

### 2.10. Statistics

All statistical analyses were done using the Prism software by GraphPad (V8). Unless otherwise noted in figure legends, statistics were obtained by performing a two-way ANOVA followed by Tukey’s multiple comparisons test.

## 3. Results

### 3.1. High-Dose shIDO-ST Treatment Induces a Dominant APC-Neutrophil Response

Therapeutic shIDO-ST treatment has been shown to induce a predominantly cytotoxic neutrophil response that is directly involved in rapid clearance of tumor cells [22,35]. To better assess the potential synergy between shIDO-ST and ICB treatment, we first utilized tumor-free C57BL/6 mice to circumvent mechanisms of suppression exerted by NSCLC tumors. Intravenous injection of therapeutic doses of shIDO-ST or scrambled shRNA control (shScr-ST), resulted in an increase of neutrophils in the bloodstream upwards of ten times higher than in vehicle-injected mice (no ST) within 48 h (Figure 1A). By 72 h, neutrophils in the bloodstream began to decrease in shScr-ST- and no-ST-treated mice but remained significantly higher (*p* < 0.022) in shIDO-ST-treated mice. The percentage of Ly6G^+^ neutrophils was also significantly elevated (*p* < 0.0007) in the spleens of mice administered shIDO-ST, compared to shScr-ST control treatment (Figure 1B). Interestingly, phenotypic analysis of splenic neutrophils from shIDO-ST-treated mice revealed dramatically increased expression of professional APC markers, which included MHCII and CD80, compared to control-treated mice. These observations are in line with literature reporting the antigen presentation capabilities of certain neutrophil populations [46].

To determine the antigen-presenting capacity of neutrophils induced by shIDO-ST treatment, a Percoll gradient was used to isolate neutrophils for in vitro stimulation assays [30,47]. To obtain the highest yield of neutrophils for this experiment, shIDO-ST or shScr-ST were injected directly into the peritoneal cavity of mice, which reliably induces a large influx of neutrophils. Separation using 4-layer Percoll gradient enabled isolation of two populations of neutrophils, low- and high-density, with the higher density neutrophils showing extreme nuclear segmentation consistent with neutrophil activation (Figure 1C) [48,49]. The predominance of low- and high-density cells isolated by Percoll gradient were Ly6G^+^ (≥90% and ≥94% purity, respectively). Purified neutrophil populations were then fixed and incubated with the ovalbumin synthetic peptide SIINFEKL, followed by incubation with splenocytes from OT-I mice transgenic for the SIINFEKL-specific T cell receptor [50]. Following co-incubation, splenocytes were analyzed for activated CD8^+^IFNγ^+^ T cells. In shIDO-ST-treated mice, the percentage of CD8^+^IFNγ^+^ T cells was significantly higher compared to shScr-ST-treated mice (Figure 1D, *p* < 0.05), indicating an enhanced ability of shIDO-ST-induced neutrophils to augment SIINFEKL-specific T cell responses. As expected, these results were only observed in the high-density (activated) neutrophil fraction. Low-density neutrophils from shIDO-ST- or shScr-ST-treated mice did not significantly induce CD8^+^IFNγ^+^ responses (Appendix A). These results are the first to report a potentially novel approach to generate functional hybrid APC-neutrophils, however, the immune response is highly skewed towards neutrophils, which may not be optimal for improving ICB efficacy [36,37]. Therefore, we next determined the effects of sub-therapeutic shIDO-ST doses in modifying the immune contexture.

### 3.2. Sub-Therapeutic shIDO-ST Treatment Circumvents Dominant Neutrophil Responses While Expanding Professional APC Subsets

For these studies, we hypothesized that significantly reducing the dose of shIDO-ST might lead to a less exaggerated neutrophil response so that other immune subsets might be preserved or augmented to improve therapeutic efficacy of ICB treatment. We first confirmed that the ST vector, used to generate shIDO-ST, could colonize lung tumors at a lower dose in the transplantable LLC1 tumor model. We engineered the ST vector to express the bioluminescent LUX reporter (LUX-ST) [51,52] in order to track movement in vivo using intravital imaging. We observed that intravenous injection of 1 × 10^6^ cfu LUX-ST [53] resulted in colonization of LLC1 tumors within 24 h (Figure 2A). Bioluminescent signal in tumors peaked by day 3 and then began to decline, with measurable levels still detectable 7 days after LUX-ST injection. To ensure that 1 × 10^6^ cfu of shIDO-ST did not have immediate anti-tumor activity as observed using high-dose shIDO-ST [22,35,54,55], we treated mice harboring LLC1 tumors with three daily doses of 1 × 10^6^ cfu shIDO-ST. This treatment regimen had no significant effect on tumor growth (Figure 2B) and did not change the frequency or APC status of tumor-associated neutrophils (Figure 2C,D). These results suggest that sub-therapeutic doses of shIDO-ST can effectively colonize LLC1 tumors without expanding a cytotoxic neutrophil response that might diminish other potential anti-tumor immune subsets.

We next examined the effects of low-dose shIDO-ST treatment on immune cell infiltration in a ***Kras^LSL-G12D/+^**; **Trp53^R270H/+^***(KP) mouse model of NSCLC, where Kras and p53 mutations can be activated in the lung by intranasal administration of adenovirus encoding Cre recombinase (Ad-cre) [41]. Following published timelines for tumor growth, we treated mice with shScr-ST or shIDO-ST (2 months post-Ad-cre administration) at low dose (1 × 10^6^ cfu) [22,35] and isolated lungs from treated groups 24 h later. H&E staining of lung sections revealed small cancer foci surrounded by hyperplastic growth of transformed lung cells (Figure 2E, H&E panels). Staining of sections with primary antibodies against CD4, CD8 and CD11c revealed that shIDO-ST treatment significantly increased infiltration of CD11c^+^ cells (*p* < 0.0001) in transformed lung tissue (as determined by IHC), while CD4^+^ and CD8^+^ cells remained unchanged and were largely confined to blood vessels with little infiltration into tumor tissue (Figure 2E). As the KP model showed inconsistent lung transformation with large variation in size of tumor foci post-administration of the Ad-cre, we chose to employ the LLC1 model for the remainder of our studies, which is the only highly reproducible lung cancer model utilizing immunocompetent mice [56].

### 3.3. Sub-Therapeutic Doses of shIDO-ST Preserves Intratumoral T Cell Frequency, Increases Antigen Presentation Potential and Reduces Treg Frequency

We next determined if sub-therapeutic shIDO-ST treatment allowed for changes in the immune landscape that could potentially augment ICB treatment. Using flow cytometry, we found that the frequencies of tumor-associated CD4^+^ and CD8^+^ T cells and myeloid-derived immune subsets 48 h post-shIDO-ST treatment were comparable to HBSS (no-ST) and shScr-ST control groups (Figure 3A, Appendix A), suggesting that these populations are preserved in the absence of a cytotoxic neutrophil response [22]. Additionally, while low-dose shIDO-ST treatment in the LLC1 model did not expand hybrid APC-neutrophils (Figure 2D), we did observe increases in antigen presentation machinery and co-stimulatory markers on CD11c^+^ dendritic cell, CD11b^+^F4/80^+^ macrophage, and CD11b^+^Ly6C^+^ monocyte populations which included increased CD86 and MHCII surface expression (Figure 3B,C). Interestingly, coincident with increases in the antigen-presenting potential of CD11c^+^ and CD11b^+^ immune subsets, sub-therapeutic shIDO-ST treatment also significantly decreased the frequency of intratumoral CD4^+^CD25^+^FoxP3^+^ Tregs (*p* = 0.0475) (Figure 3D). These results confirm that low-dose shIDO-ST treatment effectively alters the LLC1 tumor microenvironment by increasing antigen presentation potential and decreasing the presence of suppressive Tregs.

### 3.4. Sub-Therapeutic shIDO-ST Treatment Suppresses Checkpoint Protein Expression in Splenic Immune Subsets

Besides interactions within the tumor, interactions between splenic T cells and suppressive immune subsets or those expressing checkpoint proteins may inhibit T cell priming against tumor antigens in the spleen [57], or mobilization of naïve T cells from the spleen to the tumor for priming there [58]. Splenic neutrophils, monocytes, dendritic cells, and macrophages show no changes in expression of the antigen presentation molecules CD86 and MHCII (Appendix A). However, total CD45^+^ cells are significantly increased in shIDO-ST-treated mice compared to shScr-ST-treated mice (*p* < 0.0001) (Appendix A), possibly representing an increase in B cells or another subset, or the combination of small increases in each cell type, as no specific cell type assayed was significantly increased (Appendix A). Splenic Tregs were unchanged by shIDO-ST treatment (Appendix A). However, shIDO-ST treatment was capable of decreasing expression of the checkpoint molecules PD-L1, PD-1, and CTLA-4 on certain cell types. PD-L1 surface expression on CD4^+^ and CD8^+^ T cells, CD11c^+^ cells, macrophages, and NK cells is decreased significantly in shIDO-ST treated mice compared to shScr-ST treated mice (Figure 4A). In addition, shIDO-ST treatment significantly decreases surface expression of the PD-1 receptor on CD4^+^ and CD8^+^ T cells, monocytes, and NK cells (Figure 4B). Furthermore, shIDO-ST treatment significantly decreases CTLA-4 expression on CD4^+^ and CD8 T^+^ cells as well as NK cells (Figure 4C). Decreased expression of all three checkpoint proteins has the potential to globally decrease suppression, creating a TME more conducive to checkpoint blockade with both anti-PD-1 and anti-CTLA-4 antibodies.

### 3.5. ShIDO-ST Pre-Treatment Augments ICB Efficacy and Is Associated with Enhanced Immune Infiltration

Despite increased neutrophil infiltration and monocyte expression of APC-like markers in the tumor, as well as decreased numbers of suppressive Tregs in the tumor and decreased expression of checkpoint molecules in the spleen, shIDO-ST treatment is insufficient to activate intratumoral T cells. The intracellular expression of IFNγ in either CD4^+^ (Appendix A) or CD8^+^ T cells (Appendix A) remains unchanged between shScr-ST and shIDO-ST-treated groups, which may explain the lack of tumor control by sub-therapeutic doses of shIDO-ST alone (Figure 2B). However, the addition of anti-PD-1 and anti-CTLA-4 antibodies to shIDO-ST treatment (shIDO-ST + ICB) resulted in significant tumor control compared to shScr-ST + IgG or shIDO-ST + IgG as well as shScr-ST + ICB. Beginning on day 15 the growth of shIDO-ST + ICB treated tumors was significantly reduced compared to all other groups: shScr-ST + IgG (*p* = 0.0007), shIDO-ST + IgG (*p* = 0.0020) and shScr-ST + ICB (*p* = 0.0020). This reduced tumor growth remained significant through the end of the study, while tumor growth in control groups did not differ significantly throughout the study (Figure 5A). These data indicate that only shIDO-ST is capable of priming the TME to improve ICB treatment as tumors treated with shScr-ST + ICB continue to grow aggressively. We next characterized changes in the immune landscape using flow cytometry to quantify the numbers of CD45^+^ infiltrating immune cells in both the tumor and spleen. Overall CD45^+^ cells in the tumor increased significantly in the shIDO-ST + ICB group compared to shScr-ST + ICB and shIDO-ST control groups (*p* < 0.0001 for all comparisons) (Figure 5B). Total CD45^+^ cells in the spleen remained unchanged between groups (Figure 5C). Viability of CD45 negative cells in the tumor (representing tumor cells and other non-immune stromal cells) did not differ significantly between groups (consistent with the tumor growth curve), however, there is a noticeable trend in increased CD45 negative cell death in the shIDO-ST + ICB treated group (Appendix A).

A more in-depth analysis of specific immune subsets, however, revealed changes in the infiltration of multiple cell types in both the tumor and spleen. The shIDO-ST + ICB treated tumors have significantly higher numbers of tumor-infiltrating CD4^+^ and CD8^+^ T cells compared to all other treatment groups (*p* < 0.0001), as well as significantly higher numbers of CD11c^+^ cells (Figure 6A). Tumor-infiltrating CD11b^+^F4/80^+^ cells remained significantly lower in shIDO-ST + IgG and shIDO-ST + ICB groups compared to shScr-ST groups indicating that this effect is due to shIDO-ST treatment alone, whereas ICB alone decreased CD11b^+^Ly6C^+^ in both shScr and shIDO-ST groups. Tumor-infiltrating NK cells were significantly decreased only in the shIDO-ST + ICB treatment group (Figure 6A). ICB treatment in both shScr-ST and shIDO-ST groups increased Tregs out of total CD4^+^ T cells consistent with reports in human tumor treatment [59,60]; however, shIDO-ST treatment in combination with ICB prevented the level of Treg increase seen in the shScr-ST + ICB treatment group (Figure 6B). These increased Treg after ICB treatment may explain the trend in increased tumor growth in the shScr-ST + ICB treated group. CD4^+^ and CD8^+^ T cells were analyzed for expression of activation markers including CD27 (promotes T cell survival and enhances priming) [61], CD62L (rapidly lost from T cells after specific antigen stimulation) [62], granzyme B (effector, cytotoxic factor) [63], interferon gamma (IFNγ, multi-cell immune modulator secreted by activated T cells) [64] and NK1.1 (in conjunction with CD3 positivity marks natural killer T cells (NKT)) [65]. Out of CD4^+^ and CD8^+^ cells in tumors, most markers remained unchanged between shScr-ST + ICB and shIDO-ST + ICB groups except for NK1.1, which was significantly decreased in shIDO-ST + ICB treated tumors compared to shScr-ST + ICB treated tumors (CD4^+^
*p* = 0.0059, CD8^+^
*p* = 0.0027) (Figure 6C). However, due to the overall increase in CD4^+^ and CD8^+^ T cells in shIDO-ST + ICB-treated mice, these data denote a greater number of activated, intratumoral T cells compared to other groups. In the spleen, ICB treatment combined with shScr-ST decreases CD4^+^ and CD8^+^ T cells compared to IgG combined with either shScr-ST or shIDO-ST treatment; however, the combination of ICB with shIDO-ST prevents this decrease. In addition, shIDO-ST combined with ICB treatment alone increases CD11c^+^ cells and decreases CD11b^+^F4/80^+^ macrophages and CD11b^+^Ly6G^+^ neutrophils. Interestingly, shScr-ST with ICB increases the presence of CD3^-^NK1.1^+^ cells significantly in the spleen while shIDO-ST plus ICB combination prevents this increase (Figure 6D). Tregs remained unchanged in the spleen (Figure 6E) and, similar to results observed in tumor, activation markers for T cells also remained unchanged (Figure 6F).

Lastly, we determined whether various aspects of antigen presentation and intratumoral immune infiltration had any bearing on overall survival in NSCLC patients (Figure 6G). For MHCII, increased survival is especially pronounced in patient tumors with enriched CD4^+^ and CD8^+^ T cell content as well as decreased Treg content, similar to results observed using shIDO-ST+ICB. Furthermore, MHCII expression in tumors with decreased CD4^+^ and CD8^+^ T cells and increased Tregs did not correlate with changes in survival. Increased CD11c^+^ expression significantly correlated with increased overall survival in tumors with enriched CD4^+^ and CD8^+^ T cells with decreased Tregs as well as in tumors with decreased CD4^+^ and CD8^+^ T cells with increased Tregs (Figure 6F). Overall, these data suggest that multiple factors involving antigen presentation and immune infiltration will likely need to be considered when developing biomarker panels or combination strategies to improve ICB response.

## 4. Discussion

Known mechanisms of ICB resistance include the presence of immune-suppressive cell types, such as Tregs and MDSCs, as well as mechanisms that preserve T cell tolerance despite ICB. Secretion of cytokines that maintain these mechanisms of resistance can come from tumor cells as well as many immune cell types, indicating that global changes in the TME as well as antigenic re-education of T cells may be necessary for tumor-infiltrating T cells to be fully activated by ICB. In this study we explore the ability of shIDO-ST treatment [22,35,55] to affect the immune landscape both in a tumor-free setting and using a syngeneic lung cancer model in mice. In the absence of a tumor, intravenous injection of mice with high-dose shIDO-ST dramatically increases neutrophils in the blood and spleen compared to mice injected with shScr-ST. Neutrophils present in the spleen after shIDO-ST injection showed increased expression of markers consistent with professional APCs, and isolated neutrophils from shIDO-ST injected mice activated IFNγ production in T cells via antigen-specific MHC class I presentation. These results are consistent with anti-tumor, APC-like neutrophils found in patients with lung cancer [30,31,66], and may explain the additional adaptive immune response generated in previous models using high-dose shIDO-ST [22,35].

Gene expression studies of auto-reactive T cells have revealed that T cell dysfunction occurs in phases and that these various T cell sub-populations differ in their response to checkpoint blockade [67,68,69]. Fully tolerized, self T cells—which share a very similar transcriptional profile to fixed, dysfunctional, tumor-specific T cells [70]—are completely resistant to single target checkpoint blockade treatment as well as the combination of anti-CTLA-4 and anti-PD-1. Remarkably, despite down-regulation of genes related to antigen response, re-introduction of auto-antigens has been shown to reverse tolerance and sensitize T cells to checkpoint blockade [67]. These data indicate that re-establishment of the full range of immune cell activity, or induction of novel immune functions, may be necessary for enhancing efficacy of ICB treatment. In the KP orthotopic model of mouse lung cancer, shIDO-ST treatment significantly increases CD11c^+^ professional APCs. In LLC1 tumor-bearing mice, low-dose shIDO-ST treatment preserves T cell populations and monocytes experience an increase in CD86 and MHCII expression indicating that these myeloid cells, similar to neutrophils in the tumor-free model, may have greater ability to present tumor-specific antigen to tolerized T cells. Tumor Tregs are also significantly decreased by shIDO-ST treatment. In the spleen, expression of PD-1, PD-L1, and CTLA-4 are decreased in multiple immune subsets. PD-1 surface expression on T cells correlates with activation; however, high expression of PD-1 has been correlated with T cell anergy [71], similar findings have been published concerning PD-L1 and CTLA-4 on various immune subsets [72,73]. These data indicate that shIDO-ST treatment alters the TME, possibly resulting in decreased suppression and increased antigen presentation which would increase the efficacy of ICB.

The addition of antibodies blocking PD-1 and CTLA-4 to shIDO-ST treatment results in increased tumor control compared to shIDO-ST with IgG or shScr-ST with ICB treatment. This increased control may be attributed to an overall increase in the CD45^+^ fraction of tumor cells including CD4^+^ and CD8^+^ T cells, as well as CD11c^+^ cells. Notably downregulated cells types include CD11b^+^F4/80^+^ macrophages, CD11b^+^Ly6G^+^ neutrophils, and CD3-NK1.1^+^ NK cells, possibly indicating that these cell types are suppressive in the LLC1 model. Percentages of activated tumor-infiltrating T cells remained unchanged between shScr-ST + ICB and shIDO-ST + ICB groups, but the overall increase in these T cells in the shIDO-ST + ICB group indicates greater numbers of activated T cells in the tumor overall. Interestingly, NK1.1 expression on CD3^+^CD4^+^ or CD8^+^ T cells was decreased in shIDO-ST + ICB treated tumors compared to shScr-ST + ICB treated tumors, indicating fewer NKT. These cell types have previously been shown to either promote or hinder anti-tumor immunity depending on their cytokine production profile, with type I NKTs producing IFNγ to recruit NK cells and activate DCs [74], whereas Type II NKTs produce IL-13 and TGF-β that suppress anti-tumor immunity [75,76]. In tumors, suppressive type II NKTs may dominate due to the presence of Tregs [77], and ICB may increase this population due to the increased Tregs seen in Figure 6C. However, shIDO-ST + ICB treatment prevented the increase in this suppressive population despite the Treg increase.

Taken together, these data indicate that shIDO treatment alone primes the TME for treatment with ICB, and data from human lung adenocarcinoma showing that high expression of CD11c and MHCII correlate significantly with longer overall survival support the hypothesis that increased antigen presentation by unconventional or professional APCs can facilitate tumor immune surveillance and would increase the efficacy of ICB treatment. One study found that T cells newly introduced to antigen were capable of responding favorably to ICB, but T cells tolerized to self-antigen or tumor-antigen did not respond to ICB alone [67]. However, re-introduction of antigen synergized with ICB to produce functional T cells capable of eliminating tumor cells expressing self- or tumor-antigens [67]. Our data showing increased expression of antigen presentation machinery on the surface of monocytes and decreased Treg abundance in shIDO-ST treated mice may have increased antigen presentation to T cells overall, thus priming the TME for ICB treatment which only increased T cell infiltration in combination with shIDO-ST. In addition, decreasing immune checkpoint proteins on the surfaces of multiple cell types, as we have shown occurs after shIDO-ST treatment in splenocytes, may aid in increasing antigen presentation. One study showed that blocking PD-L1 with antibody on MDSCs causes a change in cytokine secretion, namely downregulation of IL-6 and IL-10 [29], which could have profound effects on the TME.

The link between IDO expression in the TME and antigen presentation is largely unknown, but some mechanisms have been described concerning the effects of agonists of the aryl hydrocarbon receptor (AhR) similar to kynurenine (the product of tryptophan conversion by IDO). In addition, to the deleterious effects of kynurenine on T cells [78,79], agonists of the AhR have been shown to regulate the immunogenicity of other cell types including APCs. Administration of the AhR agonist 2-(1’H-indole-3’-carbonyl)-thiazole-4-carboxylic acid methyl ester (ITE) to mice prevented the development of experimental autoimmune encephalomyelitis through tolerization of dendritic cells after decreasing their expression of CD86 while increasing secretion of anti-inflammatory cytokines such as IL-10 and TGF-β [80]. In vitro, ITE-treated bone marrow-derived DCs were unable to activate T cells in co-culture, and passive transfer of ITE-treated DCs were alone able to significantly inhibit experimental autoimmune encephalomyelitis [80]. In an allergy model, DCs isolated from the lungs of AhR^-/-^ mice were able to stimulate greater T cell responses to specific antigen in vitro and showed higher expression of CD86 and MHCII compared with wildtype DCs [81]. These data indicate that decreased production of AhR agonists such as kynurenine after IDO inhibition or knockdown can directly affect the antigen presentation capabilities of professional APCs such as DCs. If these pathways overlap in other immune cell types such as neutrophils or monocytes, then IDO knockdown, via shIDO-ST, may be directly responsible for upregulation of their ability to present antigen, and the overall effects of IDO knockdown may account for ICB synergy due to increased antigen presentation through multiple cell types in the LLC1 lung cancer model.

The ability of shIDO-ST to decrease the suppressive immune phenotype of the TME in the LLC1 NSCLC model, represents a promising mechanism to increase the action of ICB treatments in NSCLC patients. Recently, the CheckMate 9LA phase 3 clinical trial determined that the first line combination of nivolumab (anti-PD-1) and ipilimumab (anti-CTLA-4) with chemotherapy significantly increased overall survival in stage IV and recurrent NSCLC compared to chemotherapy alone [82]. It is a generally accepted mechanism that chemotherapy aids immunotherapy by causing tumor cell death which releases tumor-associated antigens and re-establishes T cell sensitivity [83], similar to the mechanism we have presented whereby increased antigen presentation re-establishes T cell sensitivity to ICB. Thus, clinical treatment with shIDO-ST treatment in combination with ICB may increase overall survival in patients with NSCLC or be used to increase the effectiveness of ICB with chemotherapy. The ability to increase the number of patients that will respond effectively and durably to ICB treatments will fill an unmet need within the cancer care community and may represent an avenue for increasing the efficacy of ICB in other tumor types that are as yet unresponsive.

## Figures and Tables

**Figure 1 biomedicines-08-00617-f001:**
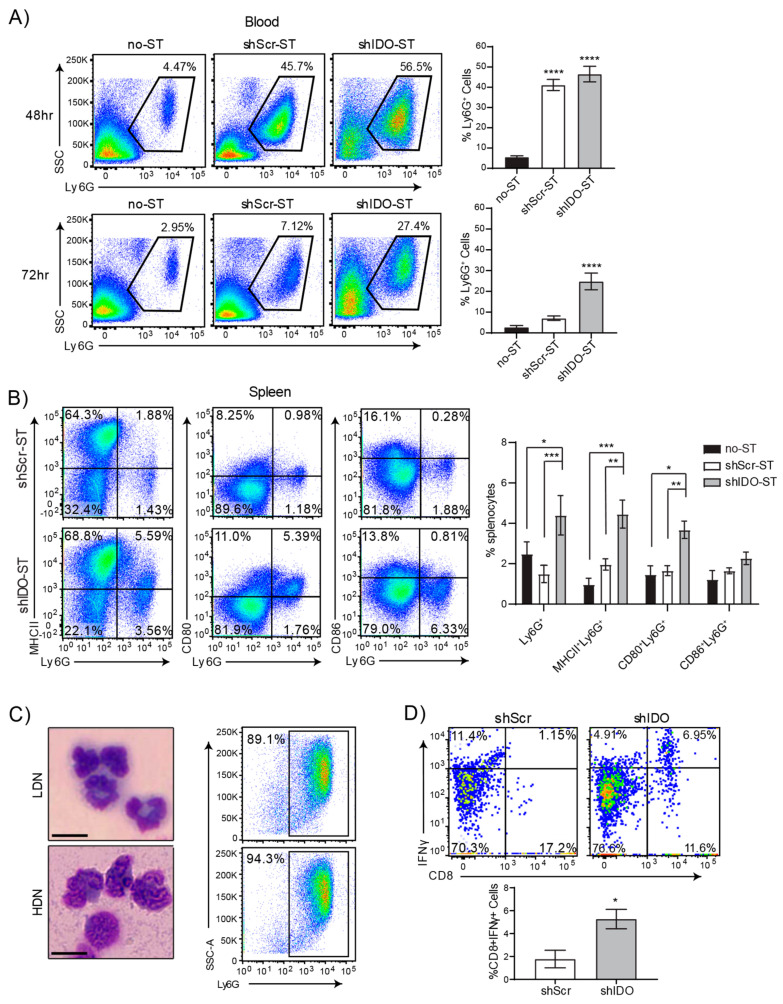
High-dose shIDO-ST treatment is associated with a neutrophil preferential response. (**A**) Flow cytometry showing neutrophil frequencies in blood by Ly6G staining out of CD45^+^ cells 48 or 72 h after treatment with HBSS (no-ST), shScr-ST or shIDO-ST. Representative gated dot plots are shown. Bar graphs show quantification of flow cytometry results. *n* = 4 mice per group. Statistics shown are compared to no-ST. (**B**) Flow cytometry showing double positivity for Ly6G and either MHCII, CD80, or CD86 for splenocytes 72 h after treatment with shScr or shIDO-ST. Representative gated dot plots are shown. Bar graphs show quantification of flow cytometry results. *n* = 4 mice per group. (**C**) Brightfield images (Wright’s stain) of cells isolated from the interfaces between 52 and 62.5% Percoll layers (LDN, low density neutrophils) and the interface between 62.5% and 78% Percoll layers (HDN, high density neutrophils). Scale bar = 10 µm. Flow cytometry dot plots represent the purity of those fractions by Ly6G positivity. (**D**) Flow cytometry of activated CD8 T cells (CD8 and IFNγ double positive) out of total OTI splenocytes after 24 h of in vitro co-incubation with SIINFEKL-loaded HDN from either shScr or shIDO-ST treated mice. Bar graph shows quantification of flow cytometry results. *n* = 3 mice per group. Unpaired *t*-test. * *p* < 0.05, ** *p* < 0.01, *** *p* < 0.001, **** *p* < 0.0001.

**Figure 2 biomedicines-08-00617-f002:**
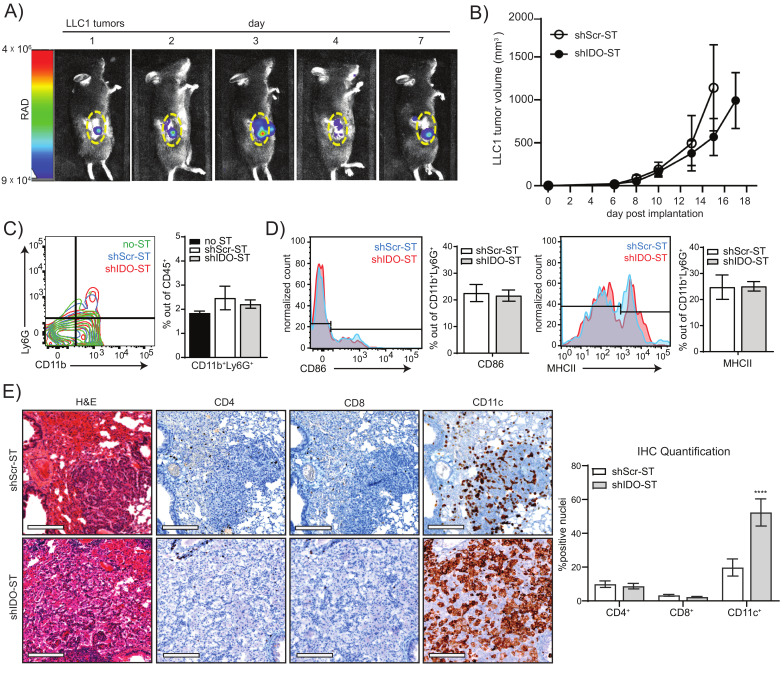
Colonization, tumor growth, and immune cell phenotypes after sub-therapeutic shIDO-ST treatment. (**A**) Mice were implanted with LLC1 tumor cells subcutaneously and allowed to grow to an average volume of 200 mm^3^. 1 × 10^6^ cfu of ST carrying a bacterial luciferase (LUX-ST) was injected intravenously and mice were imaged for bioluminescent signal on days 1, 2, 3, 4, and 7 post ST administration. Bioluminescent signal is shown in rainbow corresponding to radiance (RAD). Yellow dotted line borders the tumor. (**B**) Six days after implantation of LLC1 cells into mice, mice were treated with three consecutive, daily doses of 1 × 10^6^ cfu shScr or shIDO-ST. Tumor volumes were measured three times weekly until maximum allowed tumor growth was reached. *n* = 4 mice per group. (**C**) Flow cytometry of neutrophil frequency (CD11b^+^Ly6G^+^) out of total CD45^+^ cells from tumors treated with HBSS (no-ST), shScr-ST or shIDO-ST. Tumors were processed 48 h after the third treatment. Bar graph shows quantification of flow cytometry results. (**D**) Flow cytometry of antigen presentation molecules on neutrophils (CD86^+^ or MHCII^+^ out of CD11b^+^Ly6G^+^ cells). Bar graph shows quantification of flow cytometry results. (**E**) KP mice with lung tumors arising from oncogenic transformation were treated with 1 × 10^6^ cfu of shScr-ST or shIDO-ST. Twenty-four hours after treatment with ST, mice were euthanized and lungs were removed and fixed for H&E and IHC staining (using CD4, CD8 and CD11c antibodies). Representative images shown, scale bar = 75 µm. DAB staining for each antibody was quantified out of total hematoxylin stained nuclei for each antibody (5–12 areas per lung). Quantifications are shown as bar graphs. **** *p* < 0.0001.

**Figure 3 biomedicines-08-00617-f003:**
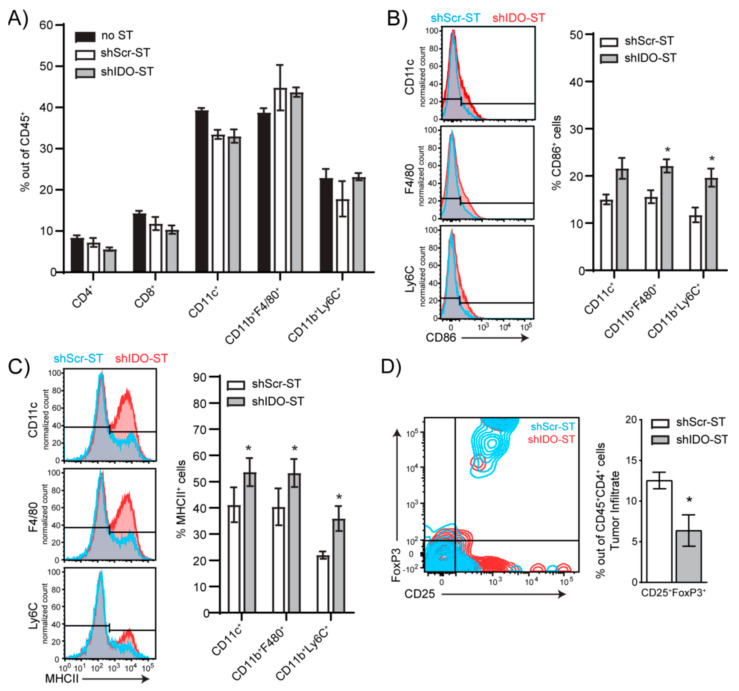
Frequency and phenotypes of tumor-infiltrating immune cells after shIDO-ST treatment. Mice (*n* = 4) with LLC1 tumors (> 80 mm^3^) were treated with three consecutive doses of 1 × 10^6^ cfu of shScr-ST or shIDO-ST and tumors were processed for flow cytometry 48 h after the third treatment. (**A**) Histograms depicting no change in frequencies of CD4^+^, CD8^+^, CD11c^+^, CD11b^+^F4/80^+^ and CD11b^+^Ly6C^+^ cells out of CD45^+^ tumor-infiltrating cells following ST treatment. (**B**) Frequency of CD86^+^ cells out of myeloid cell types (CD11c^+^, CD11b^+^F4/80^+^, and CD11b^+^Ly6C^+^ cells). Bar graphs show quantification of flow cytometry results. (**C**) Frequency of MHCII^+^ cells out of myeloid cell types (CD11c^+^, CD11b^+^F4/80^+^, and CD11b^+^Ly6C^+^ cells). Bar graphs show quantification of flow cytometry results. (**D**) Frequency of Tregs (CD25^+^FoxP3^+^ out of CD45^+^CD4^+^ cells). Bar graph shows quantification of flow cytometry results. Unpaired *t*-test. (A–D) * *p* < 0.05.

**Figure 4 biomedicines-08-00617-f004:**
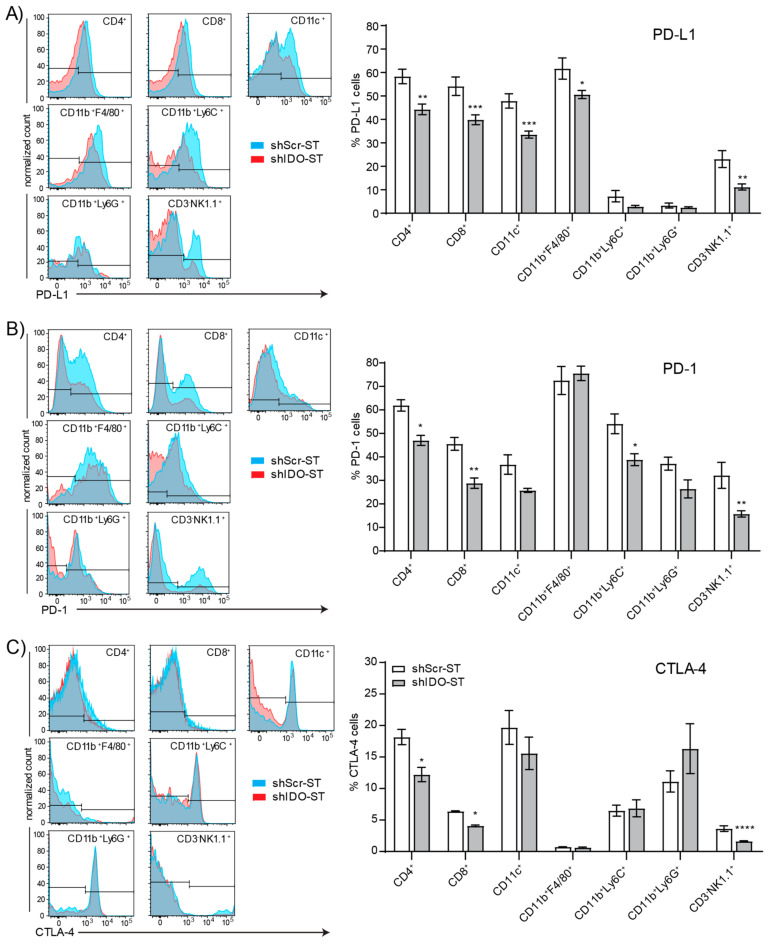
Checkpoint protein positivity of splenocytes by immune cell type after shIDO-ST treatment. Mice with LLC1 tumors (average 80 mm^3^) were treated with three consecutive doses of 1 × 10^6^ cfu of shScr-ST or shIDO-ST and spleens were processed for flow cytometry 48 h after the third treatment. *n* = 3 mice per group. Cells were gated by CD45 positivity and then by cell type (CD4^+^, CD8^+^, CD11c^+^, CD11b^+^F4/80^+^, CD11b^+^Ly6C^+^, CD11b^+^Ly6G^+^, and CD3^-^NK1.1^+^). Flow cytometry is represented by histogram overlay followed by bar graph quantification of histogram results. (**A**) Frequency of PD-L1^+^ cells by immune cell type. (**B**) Frequency of PD-1^+^ cells by immune cell type. (**C**) Frequency of CTLA-4^+^ cells by immune cell type. (A–C) * *p* < 0.05, ** *p* < 0.01, *** *p* < 0.001, **** *p* < 0.0001.

**Figure 5 biomedicines-08-00617-f005:**
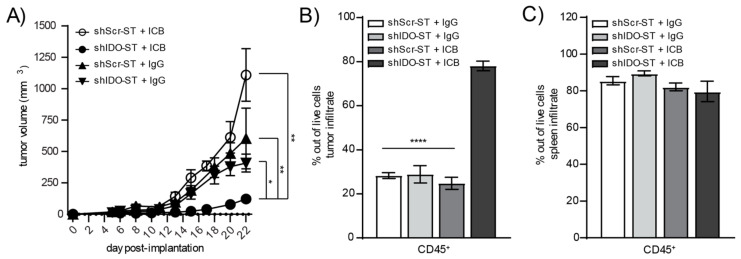
Tumor growth and total immune infiltrate after combination treatment with shScr or shIDO-ST and antibodies targeting PD-1 and CTLA-4. Mice with palpable LLC1 tumors were treated with three consecutive doses of 1 × 10^6^ cfu of shScr or shIDO-ST combined with immune checkpoint blockade (ICB, PD-1 and CTLA-4 antibodies) or IgG equivalent. PD-1 was administered at a dose of 200 µg and CTLA-4 was administered at a dose of 75 µg every three days until most groups reached maximum tumor growth. *n* = 4–10 mice per group. (**A**) Tumors were measured 3 times weekly and volume in mm^3^ is presented as a tumor growth curve. Statistical significance between shIDO-ST + ICB and shIDO-ST + IgG is shown for days 17, 20, 22, and 24 using a Mann–Whitney test. (**B**) Tumors and (**C**) spleens were excised 48 h after the third ST treatment (24 h after the second ICB or IgG treatment) and processed for flow cytometry. Frequencies of infiltrating CD45^+^ immune cells are shown in bar graphs. * *p* < 0.05, ** *p* < 0.01, **** *p* < 0.0001.

**Figure 6 biomedicines-08-00617-f006:**
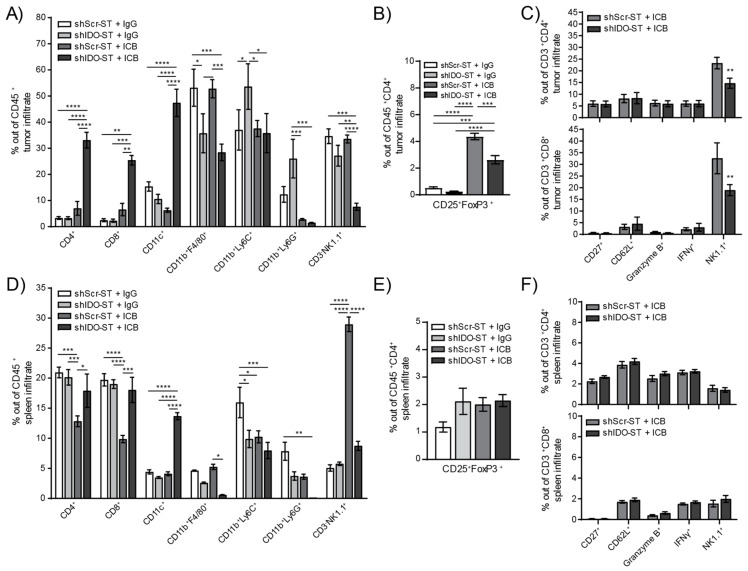
Frequency and phenotypes of tumor- and spleen-infiltrating immune cells after combination treatment with shScr or shIDO-ST and ICB. Mice with palpable LLC1 tumors were treated with three consecutive doses of 1 × 10^6^ cfu of shScr or shIDO-ST combined with immune checkpoint blockade (ICB, PD-1 and CTLA-4 antibodies) or IgG equivalent. Every three days, PD-1 was administered at a dose of 200 µg and CTLA-4 was administered at a dose of 75 µg until the end of the study. *n* = 3–4 mice per group. (**A**) Tumors were excised 48 h after the third ST treatment (24 h after the second ICB or IgG treatment) and analyzed by flow cytometry for frequency of infiltrating immune cells out of the total CD45^+^ fraction. (**B**) Tumors were also analyzed by flow cytometry for the frequency of Tregs (CD25^+^FoxP3^+^ cells out of CD45^+^CD4^+^ cells). (**C**) T cells from tumors (CD3^+^CD4^+^ or CD3^+^CD8^+^) were analyzed by flow cytometry for activation though various markers (CD27, CD62L, granzyme B, IFNγ, and NK1.1). (**D**) Spleens were excised 48 h after the third ST treatment (24 h after the second ICB or IgG treatment) and analyzed by flow cytometry for frequency of infiltrating immune cells out of the total CD45^+^ fraction. (**E**) Spleens were also analyzed by flow cytometry for the frequency of Tregs (CD25^+^FoxP3^+^ cells out of CD45^+^CD4^+^ cells). (**F**) T cells from spleens (CD3^+^CD4^+^ or CD3^+^CD8^+^) were analyzed by flow cytometry for activation using various markers (CD27, CD62L, granzyme B, IFNγ, and NK1.1). (**G**) Kaplan–Meier plots of overall survival for human adenocarcinoma correlated with high or low expression of MHCII (averaged expression of HLA-DR, HLA-DRB1, HLA-DRB5, and HLA-DRB6, top row) or CD11c (ITGAX, bottom row) genes for all available samples (CD4↑↓ CD8↑↓ Treg↑↓), for samples with enrichment of CD4 and CD8 T cells with decreased Treg cells (CD4↑ CD8↑ Treg↓), and for samples with decreased CD4 and CD8 cells with enriched Treg cells (CD4↓ CD8↓ Treg↑). * *p* < 0.05, ** *p* < 0.01, *** *p* < 0.001, **** *p* < 0.0001.

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
