# Peer review of "Salmonella-Based Therapy Targeting Indoleamine 2,3-Dioxygenase Restructures the Immune Contexture to Improve Checkpoint Blockade Efficacy"

_biomedicines, 2020, doi:10.3390/biomedicines8120617_

Round 1
Reviewer 1 Report
This article, the authors describe the application of a formerly developed Salmonella typhimurium-based platform targeting the suppressive molecule indoleamine 2,3-dioxygenase (shIDO-ST). Combination of shIDO-ST treatment and checkpoints blockage antibodies resulted in tumor regression in animal model of lung cancer. The potentially novel aspect of this study is a combination of bacterial-mediates targeted therapy and checkpoints blockage inhibitors to modulate tumor microenvironment.
There are some concerning points for this article
Major points:
- Figure 5A, shIDO-ST + IgG group (black circle) shows the best result to ameliorate the tumor. Contradictory, the author mentioned that shIDO-ST + ICB group shows the best result. Is this the error on the label in this figure?
- Compare animal experiment results from Fig 2B and Fig 5A, the data are not consistent. Both experiments were treated with shIDO-ST/shScr-ST on day 6 post-implantation. The tumor growth rate in Fig 2B is much larger than in Fig 5A. For example, the tumor size on day 17 from control groups of Fig 5A (shScr-ST + IgG) and Fig 2B (shScr-ST), the sizes of tumor are approximately 350 mm3, and > 1000 mm3 respectively.
- Fig 2B, the data of shScr-ST group on day 17 is missing.
- The authors showed the reduction of tumor growth and the evidences of of TME modulation by flow cytometric analyses. In addition, the author may show some histological analyses of apoptotic or cell death markers to support the results.
Minor point:
Page 8, line 300- Dab staining, that should be DAB.
Author Response
REVIEWER #1
Comments and Suggestions for Authors
This article, the authors describe the application of a formerly developed Salmonella typhimurium-based platform targeting the suppressive molecule indoleamine 2,3-dioxygenase (shIDO-ST). Combination of shIDO-ST treatment and checkpoints blockage antibodies resulted in tumor regression in animal model of lung cancer. The potentially novel aspect of this study is a combination of bacterial-mediates targeted therapy and checkpoints blockage inhibitors to modulate tumor microenvironment.
There are some concerning points for this article
Major points:
Point 1. Figure 5A, shIDO-ST + IgG group (black circle) shows the best result to ameliorate the tumor. Contradictory, the author mentioned that shIDO-ST + ICB group shows the best result. Is this the error on the label in this figure?
Response 1. We would like to thank the reviewer for pointing out this unfortunate error. The figure legend above the graph is mislabeled and has been fixed for Figure 5 (pg. 12; line 400) in the revised manuscript (tracked changes).
Point 2. Compare animal experiment results from Fig 2B and Fig 5A, the data are not consistent. Both experiments were treated with shIDO-ST/shScr-ST on day 6 post-implantation. The tumor growth rate in Fig 2B is much larger than in Fig 5A. For example, the tumor size on day 17 from control groups of Fig 5A (shScr-ST + IgG) and Fig 2B (shScr-ST), the sizes of tumor are approximately 350 mm3, and > 1000 mm3 respectively.
Response 2. The animal experiment in Figure 5A contained treatment with the IgG isotype control along with shScr-ST or shIDO-ST as a control for ICB antibody treatment. The addition of the IgG had some effect on tumor growth such that the growth is slower for these groups compared to shScr-ST or shIDO-ST alone (Figure 2B). Hence the need for this important control when comparing the effects of ICB treatment in the LLC1 mouse model.
Point 3. Fig 2B, the data of shScr-ST group on day 17 is missing.
Response 3. By day 15 of this experiment, tumors in the shScr-ST group had reached the maximum diameter allowed by our IACUC protocol and had to be euthanized, thus there were no tumors to measure on Day 17. Tumors in the shIDO-ST group reached maximum size by day 17 but the difference in growth kinetics was insignificant.
Point 4. The authors showed the reduction of tumor growth and the evidences of of TME modulation by flow cytometric analyses. In addition, the author may show some histological analyses of apoptotic or cell death markers to support the results.
Response 4. We agree that histological analysis of cell death in tumors would help to corroborate tumor growth control observed for the shIDO-ST+ICB treatment group, however, at the time of flow cytometric analysis, the entire tumor was used for flow cytometric analysis in order to obtain sufficient events. To address the reviewer’s concern to the best of our ability, we have included viability data obtained from the same flow cytometric data set/time point depicting the percentage of dead CD45 negative cells for each treatment group. While we are aware that the CD45 negative subset consists of more than tumor cells, there is a noticeable trend (increase) in the percentage of dead cells present in the tumor homogenate for the group treated with shIDO-ST+ICB. This data has been added to Supplemental Figure 3C and discussed in the Results section (pg. 12; line 396-399) of the revised manuscript.
Minor point:
Point 5. Page 8, line 300- Dab staining, that should be DAB.
Response 5. This error has been fixed in the figure legend for Figure 2 (pg. 8; line 311) of the revised manuscript.
Reviewer 2 Report
The manuscript entitled “Salmonella-based Therapy Targeting Indoleamine 2,3-Dioxygenase Restructures the Immune Contexture to Improve Checkpoint Blockade Efficacy” describes the method to augment the efficacy of immune checkpoint inhibitors (anti-PD-1 and anti-CTLA-4 inhibition) by immunomodulation targeting indoleamine 2,3-dioxygenase (shIDO-ST).
Considering that CheckMate227 trial has confirmed superiority of nivolumab plus ipilimumab over platinum-doublet chemotherapy in selected population of high tumor mutation burden, the attempt to improve the efficacy of anti-PD-1 and anti-CTLA-4 inhibition is interesting.
The methods and results are interesting, but too redundant and difficult to understand in some sections. I think there is room for improvement.
Major;
- The tumor-associated myeloid cells (TAMCs) used in the manuscript include tumor-associated macrophages (TAMs), myeloid-derived suppressor cells (MDSCs), dendritic cells, Tie-2 monocytes, and tumor-associated neutrophils. TAMCs are unique in their plasticity, exhibiting different responses to different signals which are easily affected by tumor microenvironment. Two major populations of TAMCs are TAMs and MDSCs, which promote angiogenesis and immunosuppression in favor of tumor progression. Therefore, the authors should describe properly the name of TAMCs which are used in the manuscript.
- I understand that lymphocytes generally lead to tumor suppression and control while neutrophils induce pro-inflammatory cytokines and chemokines that promote tumor proliferation. Then, I myself cannot fully understand the underlying mechanism of the manuscript. The authors demonstrated that the administration of shIDO-ST increases antigen-presenting capacity of neutrophils, which leads to the tumor suppression. The mechanism is based on silencing the host IDO expression leading to massive intratumoral cell death associated with neutrophil infiltration. On the other hand, sub-therapeutic shIDO-ST treatment is attempted to avoid dominant neutrophil responses, hoping to preserve other immune cells in the Result 3.2, which was later applied to the combination therapy with anti-PD-1 and anti-CTLA-4 inhibition (the Result section 3.5). The combination therapy of shIDO-ST plus anti-PD-1 and anti-CTLA-4 inhibition is considered to exert its potential against tumor cells mainly through cytotoxic lymphocytes. I want the authors to concisely explain the mechanism including the relationship between shIDO-ST induced neutrophils and increased CD8+IFNγ+ T cells in the Introduction section (not in the Results section 3.1).
- The overall description is redundant and the Results section is too long, with too many references, which is not understandable to the potential readers. I think some parts of the Results section should be well understood by the readers when moved in the Discussion section.
- I think the difference between augmentation of anti-PD-1 and anti-CTLA-4 antibody therapy by platinum doublet chemotherapy (CheckMate 9LA) and the sub-therapeutic shIDO-ST (current study) should be discussed in the Discussion section.
Minor;
- The ground on which the anti-PD-1 and anti-CTLA-4 antibody dose should be mentioned, like shIDO-ST.
- Why does shIDO-ST induce higher immunomodulation with intraperitoneal administration compared to intravenous administration?
Author Response
REVIEWER #2
Comments and Suggestions for Authors
The manuscript entitled “Salmonella-based Therapy Targeting Indoleamine 2,3-Dioxygenase Restructures the Immune Contexture to Improve Checkpoint Blockade Efficacy” describes the method to augment the efficacy of immune checkpoint inhibitors (anti-PD-1 and anti-CTLA-4 inhibition) by immunomodulation targeting indoleamine 2,3-dioxygenase (shIDO-ST).
Considering that CheckMate227 trial has confirmed superiority of nivolumab plus ipilimumab over platinum-doublet chemotherapy in selected population of high tumor mutation burden, the attempt to improve the efficacy of anti-PD-1 and anti-CTLA-4 inhibition is interesting.
The methods and results are interesting, but too redundant and difficult to understand in some sections. I think there is room for improvement.
We would like to thank the reviewer for their thoughtful comments. Please see our following revisions in the revised manuscript with tracked changes.
Major points:
Point 1. The tumor-associated myeloid cells (TAMCs) used in the manuscript include tumor-associated macrophages (TAMs), myeloid-derived suppressor cells (MDSCs), dendritic cells, Tie-2 monocytes, and tumor-associated neutrophils. TAMCs are unique in their plasticity, exhibiting different responses to different signals which are easily affected by tumor microenvironment. Two major populations of TAMCs are TAMs and MDSCs, which promote angiogenesis and immunosuppression in favor of tumor progression. Therefore, the authors should describe properly the name of TAMCs which are used in the manuscript.
Response 1. In order to address this comment, we have made several changes to our Introduction section. We have removed all references to “tumor-associated myeloid cells (TAMCs)” and clarified the specific cell types being addressed.
Pg. 2; line 77: “anti-tumor myeloid cells” has been changed to “anti-tumor neutrophils”
Pg. 2; line 81: “(TAMCs)” has been removed
Pg. 2; line 88: “TAMCs” has been replaced with “tumor-infiltrating neutrophils”
Pg. 3; line 97-98: “TAMCs” has been replaced with “dendritic cells, macrophages and monocytes”
Point 2. I understand that lymphocytes generally lead to tumor suppression and control while neutrophils induce pro-inflammatory cytokines and chemokines that promote tumor proliferation. Then, I myself cannot fully understand the underlying mechanism of the manuscript. The authors demonstrated that the administration of shIDO-ST increases antigen-presenting capacity of neutrophils, which leads to the tumor suppression. The mechanism is based on silencing the host IDO expression leading to massive intratumoral cell death associated with neutrophil infiltration. On the other hand, sub-therapeutic shIDO-ST treatment is attempted to avoid dominant neutrophil responses, hoping to preserve other immune cells in the Result 3.2, which was later applied to the combination therapy with anti-PD-1 and anti-CTLA-4 inhibition (the Result section 3.5). The combination therapy of shIDO-ST plus anti-PD-1 and anti-CTLA-4 inhibition is considered to exert its potential against tumor cells mainly through cytotoxic lymphocytes. I want the authors to concisely explain the mechanism including the relationship between shIDO-ST induced neutrophils and increased CD8+IFNγ+ T cells in the Introduction section (not in the Results section 3.1).
The overall description is redundant and the Results section is too long, with too many references, which is not understandable to the potential readers. I think some parts of the Results section should be well understood by the readers when moved in the Discussion section.
Response 2. We appreciate this comment and the reviewer’s careful understanding of our manuscript. In order to clarify our mechanism and the relationship between the various data we have added a more detailed summary to the last paragraph of the introduction.
Pg. 2-3, lines 86-100 have been modified to read:
“Utilizing various cancer models we have confirmed that our Salmonella typhimurium-based platform targeting IDO (shIDO-ST) is engulfed by tumor-infiltrating neutrophils, leading to hyperactivation of their cytotoxic activity and subsequent tumor regression [22, 35]. In this study, we hypothesized that shIDO-ST treatment might improve the efficacy of ICB treatment in NSCLC if a greater breadth of anti-tumor immune responses could be generated including increasing antigen presentation by various immune subsets. While we confirmed the ability of high-dose shIDO-ST treatment to produce neutrophils with APC function, the overall immune response was highly skewed to this cytotoxic subset and not conducive for improving ICB efficacy [41, 42]. Alternatively, we discovered that sub-therapeutic doses of shIDO-ST could prevent the dominant neutrophil response and augment antigen presentation in professional and non-conventional APCs including dendritic cells, macrophages, and monocytes. Additionally, low-dose shIDO-ST treatment reduced the frequency of suppressive Tregs and expression of immune checkpoint molecules, ultimately improving efficacy of ICB therapy in an aggressive model of NSCLC. Our ability to demonstrate synergy between shIDO-ST and ICB treatment may support their combination as a strategy to enhance response rates in NSCLC patients [43, 44].”
Furthermore, to decrease redundancy between the results section and introduction/discussion we have made the following changes in the revised manuscript:
Previously pg. 5; line 222-223 has now been moved to the Discussion (pg.15 line 488-489).
Previously pg. 7; line 263-265 has been removed for redundancy
Previously pg. 9; line 325-328 has been removed for redundancy.
Previously pg. 10; line 362-365 has been moved to the Discussion (pg.15 line 504-507).
Point 3. I think the difference between augmentation of anti-PD-1 and anti-CTLA-4 antibody therapy by platinum doublet chemotherapy (CheckMate 9LA) and the sub-therapeutic shIDO-ST (current study) should be discussed in the Discussion section.
Response 3. As recommended by the reviewer, we have added a discussion of the CheckMate 9LA study in relation to our results on pg. 16; line 562-570 of the Discussion.
Minor points:
Point 4. The ground on which the anti-PD-1 and anti-CTLA-4 antibody dose should be mentioned, like shIDO-ST.
Response 4. Dose per mouse for anti-PD-1 and anti-CTLA-4 antibodies were determined from previous publications that typically utilize 200µg-250µg anti-PD-1 per mouse (Salazar, et al. 2016; Bertrand, et al. 2017) and 60µg-200µg anti-CTLA-4 per mouse (Van Hooren, et al. 2017; Son, et al. 2017). We chose to use doses at the lower end of those published in order to differentiate synergistic effects between shIDO-ST and ICB treatment.
This information has now been added to pg. 4; line 168-170 of the Experimental Section.
Point 5. Why does shIDO-ST induce higher immunomodulation with intraperitoneal administration compared to intravenous administration?
Response 5. In regards to immunomodulation by shIDO-ST administrated intraperitoneal versus intravenous, we assume that the function of APC-like neutrophils generated by intraperitoneal administration do not differ significantly from those isolated from blood after intravenous administration based on the results observed in Figure 1. Additionally, neutrophils from blood were difficult to isolate to pure fractions in order to perform the in vitro activation assays with OT1 splenocytes (Figure 1). The abundance of RBCs overwhelmed the closely dense Percoll layers which did not allow for proper separation. As other methods of neutrophil isolation (e.g. antibody selection) are known to cause neutrophil lysis or false activation, we utilized the neutrophils isolated after peritoneal administration for this assay which were easily purified by Percoll gradient, preserving neutrophil viability and activation state post-shIDO-ST treatment.
Round 2
Reviewer 2 Report
The manuscript is well revised and has become more understandable to the potential readers.